# The Potential for Targeting G_2_/M Cell Cycle Checkpoint Kinases in Enhancing the Efficacy of Radiotherapy

**DOI:** 10.3390/cancers16173016

**Published:** 2024-08-29

**Authors:** Emma Melia, Jason L. Parsons

**Affiliations:** 1Institute of Cancer and Genomic Sciences, University of Birmingham, Edgbaston, Birmingham B15 2TT, UK; exm307@student.bham.ac.uk; 2School of Physics and Astronomy, University of Birmingham, Edgbaston, Birmingham B15 2TT, UK

**Keywords:** carbon ions, cell cycle checkpoint, Chk1, DNA damage, DNA repair, ionising radiation, proton beam therapy, radiotherapy, Wee1

## Abstract

**Simple Summary:**

Around 50% of all human cancers are treated with radiotherapy. The effectiveness of radiotherapy is driven through causing DNA damage within the cancer cells; however, the cells respond by activating repair mechanisms that can lead to resistance to treatment. A promising strategy is to target one of these defence mechanisms, called the cell cycle checkpoint, using specific drugs/inhibitors that can be used in combination with radiotherapy. Here, we review evidence of investigations into inhibitors of two important proteins (Chk1 and Wee1) and how these can be used to increase the effectiveness of radiotherapy in cancer treatment.

**Abstract:**

Radiotherapy is one of the main cancer treatments being used for ~50% of all cancer patients. Conventional radiotherapy typically utilises X-rays (photons); however, there is increasing use of particle beam therapy (PBT), such as protons and carbon ions. This is because PBT elicits significant benefits through more precise dose delivery to the cancer than X-rays, but also due to the increases in linear energy transfer (LET) that lead to more enhanced biological effectiveness. Despite the radiotherapy type, the introduction of DNA damage ultimately drives the therapeutic response through stimulating cancer cell death. To combat this, cells harbour cell cycle checkpoints that enables time for efficient DNA damage repair. Interestingly, cancer cells frequently have mutations in key genes such as TP53 and ATM that drive the G_1_/S checkpoint, whereas the G_2_/M checkpoint driven through ATR, Chk1 and Wee1 remains intact. Therefore, targeting the G_2_/M checkpoint through specific inhibitors is considered an important strategy for enhancing the efficacy of radiotherapy. In this review, we focus on inhibitors of Chk1 and Wee1 kinases and present the current biological evidence supporting their utility as radiosensitisers with different radiotherapy modalities, as well as clinical trials that have and are investigating their potential for cancer patient benefit.

## 1. Introduction

Radiotherapy using ionising radiation remains the cornerstone of cancer treatment. This is highlighted by the fact that ~50% of all cancer patients will receive some form of radiotherapy as part of their treatment plan, which can be alone or as a combination with either chemotherapy and/or surgery. The most conventional form of radiotherapy given is in the form of X-rays (photons). However, due to the relatively high energy deposition upon entry into the body and the subsequent further release of energy along the radiation track and beyond the cancer target, this serves to create adverse treatment side effects through the high levels of damage occurring to the surrounding healthy tissues and organs at risk. In contrast, particle beam therapy (PBT) using particles such as protons and carbon ions benefits from defined dose deposition in a narrow and well-defined range that can be specifically targeted at the tumour. This is achieved through the Bragg peak that displays a low entrance and exit dose, which can subsequently spare the surrounding healthy tissues [1]. Larger cancers can be targeted with particle beams of different initial energies, creating a spread-out Bragg peak which is largely utilised clinically. Both photons and protons are considered to be low linear energy transfer (LET) radiation as the frequency of ionisation events and therefore the ensuing damage created along the radiation track are relatively sparse. However, the LET of protons increases at and around the Bragg peak and is highest at the distal end. This creates biological and clinical uncertainty with treatment of patients with proton beam therapy. This is partially reflected in the relative biological effectiveness of 1.1 of protons which is used in treatment planning, although this has been highly debated [2,3]. Other forms of PBT include carbon and helium ions, which have a considerably higher LET than photons and protons. Similarly, boron neutron capture therapy (BNCT) is an alternative radiotherapy technique which utilises thermal neutron radiation and tumours pre-treated with boron-containing compounds to create high-LET helium and lithium ions. Generally, these high-LET radiotherapies are considered important for radioresistant or recurrent tumours, including those of the head and neck and brain [4,5,6], although we have yet to fully exploit their therapeutic and clinical potential.

In addition to radiotherapy type, there have also been recent advancements in dose delivery in order to combat the adverse side effects of the treatment, particularly following X-ray radiotherapy. These include the use of ultra-high dose rates (FLASH) and spatially fractionated radiotherapy through mini/microbeams [7,8]. Preclinical evidence is accumulating to demonstrate their healthy tissue-sparing properties whilst still retaining tumour control, although there are still significant gaps in our knowledge of how these biological effects are achieved; such knowledge is vital before they are moved into the clinic for patient treatment. Despite the therapeutic benefits of radiotherapy, it is still considered important to combine this with targeted drugs/inhibitors in order to enhance the efficacy of the radiation, particularly in hard-to-treat cancers. Radiotherapy relies heavily on the introduction of DNA damage to overwhelm the cancer cells’ ability to repair this damage, ultimately resulting in cell death. Therefore, proteins within the cellular defence mechanisms are deemed key targets for inhibition and improving the biological effectiveness of radiotherapy treatment.

## 2. The Cellular DNA Damage Response (DDR)

In response to radiotherapy, a number of different lesions are generated in DNA along the radiation track [9,10], although the formation of DNA double strand breaks (DSB) and complex DNA damage (CDD) containing two or more lesions within 1–2 helical turns of the DNA [11] are considered the most lethal to the cell. CDD in particular is a signature of ionising radiation treatment. In response to this, human cells activate the cellular DNA damage response (DDR) that detects, signals and repairs the DNA lesions (Figure 1) [12]. The DDR also includes stimulating cell cycle arrest response. The G_1_/S checkpoint is triggered by recruitment and initial activation of ataxia telangiectasia mutated (ATM) at DSB sites, as well as amplification and sustained activation of ATM via the MRE11-Rad50-NBS1 (MRN) complex [13]. Activated ATM then stimulates Chk2, p53 and p21, which are mainly responsible for coordinating stalling at the G_1_/S checkpoint (Figure 1). Intra-S and G_2_/M checkpoint arrest is mainly regulated by ataxia telangiectasia and Rad3-related (ATR), Chk1 and Wee1, following DNA end resection and prior to homologous recombination (HR) repair (discussed in Section 3). Cell cycle arrest allows for efficient repair of DNA damage prior to continuation of cell proliferation. There are two different mechanisms that DSBs are generally repaired by, although pathway choice depends on multiple factors, including cell cycle stage. Sites of DSBs are often flagged by phosphorylation of histone variant H2AX (also known as γH2AX) by members of the phosphoinositide 3-kinase (PI3K)-related kinase (PIKK) family, specifically ATM, ATR and DNA-dependent protein kinase catalytic subunit (DNA-Pkcs). This phosphorylation and the resultant ubiquitination by ring-finger protein- (RNF) 8, 168, and ubiquitin-conjugating enzyme 13 (UBC13) complex result in the activation and recruitment of various DNA damage sensing agents, including 53BP1 and BRCA1. Both 53BP1 and BRCA1 are mutually antagonistic, with 53BP1 favouring non-homologous end-joining (NHEJ) and BRCA1 promoting more accurate HR repair in S and G_2_ phases [14].

Cells in G_0_/G_1_ generally utilise NHEJ, which can be divided into two sub-pathways, namely, classical NHEJ (c-NHEJ) and alternative NHEJ (a-NHEJ; Figure 1) [15]. In the c-NHEJ pathway, the Ku70/80 heterodimer binds to the DSB ends followed by recruitment of DNA-Pkcs and factors such as Artemis and polynucleotide kinase phosphatase (PNKP) that process the DNA ends. Ligation of the DSB is then achieved by the complex of X-ray cross-complementing protein 4 (XRCC4), DNA ligase IV, and XRCC4-like factor (XLF). For the a-NHEJ pathway, limited DNA end resection is stimulated by the MRN complex in concert with C-terminal-binding protein interacting protein (CtIP). Poly(ADP-ribose) polymerase-1 (PARP-1) subsequently binds the DSBs ends along with DNA polymerase θ (Pol θ) that synthesises the DNA, and lastly the DNA is ligated through the actions of either DNA ligase I or the complex of X-ray cross-complementing protein 1 (XRCC1) and DNA ligase IIIα. It is known that cells have a profound G_2_/M arrest following irradiation, of which Chk1 and Wee1 are crucial regulators via influences on CDK1 (as discussed in Section 3).

For cells in both S and G_2_, where a sister chromatid is available to be used as a template for repair, HR is largely used as an error-free mechanism to promote DSB repair [16]. The process is initiated by the MRN complex and CtIP which initiates DNA end resection, although this is greatly facilitated by EXO1, Dna2, BLM and WRN which generates 3′ single-stranded DNA (ssDNA) overhangs. These overhangs are initially coated with replication protein A (RPA) and then replaced by the recombination protein RAD51 via the actions of BRCA1-BARD1, PALB2 and BRCA2. The nucleoprotein filament formed then undergoes homology search and invasion into the sister chromatid, facilitated by RAD54, where DNA polymerases δ or ε perform DNA synthesis. DNA ligation subsequently generates a Holliday junction that is processed by resolvases, including MUS81-EME1/2, GEN1 and BLM-TopoIIIα-RMI1, that complete the DSB repair process.

CDD is uniquely formed by the track structure of ionising radiation and can consist of a variety of DNA lesions, including DSBs, single strand breaks (SSBs) and DNA base damage. The higher the LET of the radiotherapy, the greater the frequency and complexity of the damage that drives the therapeutic effect in cancer cell killing. CDD sites are extremely difficult for the cell to repair and can persist for a significant amount of time post-irradiation. Given the varied nature of the damage within CDD sites, it is likely that multiple DNA repair pathways are required to resolve this, including proteins involved in both DSB and SSB repair mechanisms [11]. In addition to DNA repair, cell cycle regulation plays an important role in the cellular DDR to enable sufficient time for the repair of the DNA damage and prevent this from disrupting DNA replication and reducing the likelihood of introducing mutations into the DNA. Although the G_1_/S checkpoint is effective, the G_2_/M checkpoint is considered more crucial because it allows for the accurate repair of DSBs via HR. Additionally, cancerous cells are highly reliant on the activity of ATR-Chk1-Wee1 at the G_2_/M checkpoint due to common oncogenic mutations in ATM and TP53.

## 3. Roles of Chk1 and Wee1

ATR is a member of the PIKK family and is crucial in the DDR. ATR is activated following the detection of ssDNA, either in the form of stalled replication forks, SSBs, or ssDNA occurring as an intermediate in the initial end resection processing during HR (Figure 1). ssDNA is usually coated in RPA, which allows for the recruitment of ATR via binding with ATR-interacting protein (ATRIP) [17,18]. The complete activation of ATR requires binding and interaction with various proteins, such as TOPBP1, ETAA1, and RAD17/9-1-1 complexes [19,20,21,22,23]. The majority of the direct influence of ATR is primarily confined to S phase in coordinating DNA replication and protecting subsequent replication forks. However, the cell cycle control aspect of ATR functioning is a result of the activation of Chk1 (Figure 2).

Phosphorylation of Chk1 on serine residues 317 and 345 by ATR is dependent on the interaction with Claspin adaptor protein, following the detection of ssDNA [24,25,26]. This modification of Chk1 results in autophosphorylation at serine 296, ultimately resulting in its kinase activation and a plethora of downstream reactions, including influences on replication stress, cell cycle control and DNA repair. Activated Chk1 directly phosphorylates Cdc25a, targeting the phosphatase for ubiquitination and ultimately proteasomal degradation [27,28]. In an unperturbed cell cycle, Cdc25a is responsible for dephosphorylating CDK2, which allows progression at G_1_/S and intra-S checkpoints [29,30]. Therefore, although this is mainly regulated by the ATM/Chk2/p53 pathway, Chk1 can influence G_1_/S arrest; however, a more important role of Chk1 results in slowed S phase in the presence of replication stress. Chk1 is also able to influence the phosphorylation of Cdc7, a key kinase involved in DNA replication initiation, influencing the loading of Cdc45 at DNA replicative origins via Mcm2-7 complexes with CDK2 activity [31]. Furthermore, Chk1 activity phosphorylates Tlk1, which prevents proficient chromatin assembly [32]. Together, these downstream targets of Chk1 result in control and stability of DNA replication, which is further implicated by the requirement of Chk1 activity for the ubiquitination of PCNA [33], a key regulator of translesion DNA synthesis.

In addition to its key regulatory function at the intra-S phase of the cell cycle and DNA replication, Chk1 has major roles in mitotic progression. Activated Chk1 phosphorylates Cdc25c, resulting in nuclear exclusion of the phosphatase along with Cdc25a, which is responsible for dephosphorylating CDK1, ultimately resulting in stalled progression at the G_2_/M checkpoint [34,35]. Arrest at this checkpoint allows for DNA damage repair, specifically via HR, and restoration of genomic integrity prior to mitosis. Importantly, for HR proficiency, it is known that Chk1 directly phosphorylates RAD51 at threonine residue 309, which influences the stability and regulation of RAD51, ultimately influencing its ability to bind chromatin, an essential step in HR [36]. The final checkpoint in the cell cycle is also influenced by Chk1 activity, due to the phosphorylation of Aurora B [37], which influences the activity and localisation of BubR1, a major kinase in the mitotic spindle assembly checkpoint (SAC). Finally, in addition to directly regulating key mediators of the cell cycle, Chk1 also phosphorylates histone H3 at threonine residue 11, which results in transcriptional repression of cell cycle regulatory genes, including various CDK’s and cyclins, due to a reduction in acetylation [38].

The activity of Wee1 is also directly influenced by Chk1. Wee1 is one of three Wee1 family kinases, along with MYT1 and Wee1B (also known as Wee2), of which Wee1 and MYT1 are specifically important in cell cycle regulation. Activated Wee1 is able to phosphorylate CDK1/2 at tyrosine residue 15 [39,40], resulting in intra-S and G_2_/M phase arrest, particularly important in both replication stress and DNA damage repair via HR (Figure 2). A major role of Wee1 is in the maintenance of genomic integrity via the direct influence on CDK1 involved in the regulation of MUS81-EME1/2 complex formation [41], crucial in the removal of branched DNA structures created through replication forks and HR. Importantly, there is also evidence for the direct interaction and regulation of MUS81 by Wee1 [42]. This regulation of MUS81-EME1/2 activity prevents the formation of DSBs and chromosomal pulverisation, thus maintaining genomic integrity. In addition, Wee1-mediated suppression of CDK2 activity can limit fork degradation by Dna2 [43]. Suppression of CDK activity by Wee1 has additionally been demonstrated to protect genome integrity by controlling replication origin firing and reducing nucleotide depletion [44]. Furthermore, Wee1 is able to directly phosphorylate histone H2B at tyrosine 37, located upstream of the histone gene cluster *Hist1*, resulting in the loss of acetylation groups and therefore influencing histone production and ultimately controlling DNA/histone dynamics [45]. Following effective DNA damage repair and the onset of mitosis, Wee1 is negatively regulated by CDK1 via phosphorylation on serine 123. This provides a docking site for PLK1 and CK1/2, further phosphorylating Wee1 at serines 53 and 121, respectively, targeting the protein for degradation via the SCFβ-TrCP ubiquitin–ligase complex [46,47,48,49]. Finally, Wee1 can directly regulate components of the E3 ubiquitin ligase complex APC/C, influencing the efficacy of the SAC [50], subsequently preventing metaphase progression in the presence of DNA damage in mitosis and aiding restoration of genomic integrity prior to finalised cellular division.

## 4. Targeting Cell Cycle Checkpoint Kinases

As a result of the aforementioned functions, both Chk1 and Wee1 are attractive targets for enhancing tumour radiosensitivity. Given their roles in intra-S, G_2_/M and SAC checkpoints in response to replication stress and DNA damage, this is given more importance as the majority of tumours have an inherent lack of active G_1_/S checkpoint activation as a result of ATM or TP53 mutations. Therefore, a complete lack of DNA damage checkpoint activation may cause cells to progress through to mitosis, with unrepaired DNA damage resulting in increased genomic instability and subsequent cell death. The preclinical observations of the underlying mechanisms of the enhanced radiosensitivity following targeting both Chk1 and Wee1 in combination with X-rays, as well as other forms of radiation with a higher LET, are discussed below, along with the current clinical progression of this combinatorial therapy.

### 4.1. Development and Progression of Chk1 Inhibitors

The first generation Chk1 inhibitor UCN-01 was originally identified as a protein kinase C (PKC) inhibitor, derived from Staurosporine in 1987 [51]. UCN-01 was first used clinically in two studies in the United States and Japan [52]; however, it is highly unspecific, targeting a range of protein kinases including CDK2, Chk1, Chk2, PKA and PKC. In 2007, the identification of the first orally available Chk1-targeting inhibitor XL-844 (EXEL-9844) was discovered, although this was a dual inhibitor of both Chk1 and Chk2 [53]. In the same year, there was also the development of CBP-501; however, this was also shown to target MAPKAP-K2 and C-Tak-1 with high affinity [54]. Nevertheless, CBP-501 progressed into clinical trials showing promising results in combination with cisplatin, specifically in ovarian and mesothelioma patients [55]. However, in further phase II trials in mesothelioma patients, CBP-501 was determined to have no improved efficacy in patients compared to standard cisplatin treatments [56].

In 2008, there was the development of AZD7762, which was a potent dual inhibitor of Chk1/Chk2, [57,58]. AZD7762 entered clinical trials for enhancing solid tumour sensitivity to gemcitabine; however, the clinical development was terminated following unpredictable cardiac toxicity [59]. PF-00477736 was developed in the same year and shown to have a 100-fold selectivity for Chk1 over Chk2 and with high specificity compared to other protein kinases [60]. PF-00477736 demonstrated promising results in Phase I clinical trials of advanced solid tumours in combination with gemcitabine (NCT00437203) [61]; however, the inhibitor was then discontinued by Pfizer. V158411, a dual Chk1/Chk2 inhibitor, has shown enhanced efficacy with DNA damage-inducing agents, such as gemcitabine and cisplatin, with no additional systemic toxicities in vivo; however, it has not progressed into clinical development [62]. Additionally, CHIR-124 was developed, which showed high potency and selectivity for Chk1 inhibition, with 500–5000 fold selectivity over other kinases tested, including Chk2 [63]. Despite this high specificity, CHIR-124 has not progressed into clinical development, similar to SAR-020106 developed in 2010 [64]. CCT244747 and CCT245737 (also known as SRA737) were developed in 2012 and were the first highly selective orally available Chk1 inhibitors [65,66]. SRA737 has recently progressed into clinical trials, showing tolerability alone [67] and progressing in combination with gemcitabine in advanced solid tumours [68]. The Chk1 inhibitors developed by Array BioPharma AR323 and AR678 have shown good single agent effects [69,70] but have not been explored as combinatorial therapies, nor have they progressed into clinical trials. Furthermore, SCH900776, later renamed and referred to herein as MK-8776, was developed and is one of the most selective and potent Chk1 inhibitors with 500-fold selectivity for Chk1 over Chk2 [71]. MK-8776 has shown promising preclinical evidence for enhancing radiotherapy efficacy (see Section 4.3) and also clinical progression in combination with chemotherapeutics such as gemcitabine and cytarabine [72,73]. Genentech has developed various Chk1 inhibitors, including GNE-783, GNE-900 [74], and more recently GDC-0425 and GDC-0575 [75,76]. The latter two were involved in phase I clinical trials alone and in combination with gemcitabine in refractory solid tumours; however, dose-limiting toxicities were observed [77,78]. LY2603618 (commercially known as rabusertib) was developed in 2014 with 1500-fold selectivity over Chk2 and 100-fold selectivity over 51 other protein kinases tested [79]. Interestingly, in clinical trials, no significant advances over conventional regimens when combined with gemcitabine or pemetrexed were seen [80,81,82]. Later, Eli Lily also developed LY2606368 (commercially known was prexasertib), a dual Chk1/Chk2 inhibitor, only showing 8-fold selectivity for Chk1 over Chk2 [83]. In 2019, despite many promising completed and ongoing clinical trials, Eli Lily announced an end to the clinical development of LY2606368.

### 4.2. Development and Progression of Wee1 Inhibitors

In comparison to Chk1 inhibitors, the development of Wee1 inhibitors has been considerably slower. The first Wee1 inhibitor PD0166285 was developed in 2001 but was shown to have low specificity and high potency for Myt1 and to target other protein kinases [84]. Nevertheless, PD0166285 initially demonstrated that Wee1 was a promising new target for increasing radiosensitisation in a variety of different tumour cell types [85]. High-throughput screening of Staurosporine also highlighted PD0407824 as a potential Wee1 inhibitor [86], from which Wee1 inhibitor II would later be derived, although due to solubility issues, there were limited applications of this inhibitor [87]. In 2019, a more potent selective Wee1 inhibitor MK-1775 (also known as AZD1775 and commercially as adavosertib) was developed [88]. MK-1775 was the first Wee1 inhibitor that progressed to clinical trials, showing a plethora of preclinical evidence for single agent and combinatorial therapy enhancements in tumour treatments. Whilst it has been shown to effectively enhance treatment of various brain tumours, there were major concerns regarding the ability of the drug to pass the blood–brain barrier [89]. However, a phase 0 trial reported adequate brain tumour penetrance, but this could be due to a leaky blood–brain barrier known to be at the tumour site [90]. Despite the major clinical efforts and applications of MK-1775, the main concern is its tolerability.

Therefore, there have been recent attempts to develop more selective and less cytotoxic Wee1 inhibitors. These include the highly potent and selective Wee1 inhibitor, Zn-c3 [91], which is currently in multiple phase I/II clinical trials either as a monotherapy or in combination with chemotherapeutic drugs. However, there is no current evidence for the radiosensitisation potential of this new Wee1 inhibitor. Another recently developed Wee1 inhibitor is IMP7068, which is also in a phase I clinical trial as a monotherapy to determine dose escalation and pharmacokinetics (NCT04768868). IMP7068 has shown promising preclinical data for tumour inhibition and tolerability in vivo [92]. Additionally, Debio-0123 has been identified as a potent Wee1 inhibitor, which is currently in dose escalation clinical trials (NCT05109975). Interestingly, Debio-0123 is in clinical trials in combination with standard treatment regimens, including radiotherapy for glioblastoma (GBM) patients (NCT05765812), as preclinical evidence has demonstrated its enhanced ability to cross the blood–brain barrier compared to previous Wee1 inhibitors [93]. Finally, new potent Wee1 inhibitors SY-4835 and SC0191 are both in phase I clinical trials, either as monotherapies or in combination with chemotherapy (NCT05291182, NCT6055348 and NCT06363552), although the combination with radiotherapy has yet to be explored. Additionally, analogues of MK-1775 have been explored preclinically and have revealed similar potency for Wee1, with significantly reduced toxicities, also demonstrating synergy with cisplatin [94], which could make them a promising alternative to MK-1775 to combat some tolerability issues.

Despite various Chk1 and Wee1 inhibitors being developed and a variety of these progressing into clinical trials, the remainder of our review will focus on the utility of these inhibitors in preclinical and clinical studies in combination with radiotherapy only.

### 4.3. Chk1 Inhibitor Preclinical Studies

Ionising radiation results in a profound arrest at the G_2_/M checkpoint orchestrated by Chk1 to enable DNA damage repair. Therefore, targeting Chk1 can significantly reduce cell stalling and is therefore thought to prevent the efficient resolution of radiation-induced DNA damage (summarised in Table 1). This is supported by a study performed in triple-negative breast cancer (TNBC) models (MB-231, BT-549 and Cal-51) which all showed significantly reduced survival following pre-treatment with the Chk1 inhibitor MK-8776, compared to the irradiated controls [95]. This sensitivity correlated with an increase in γH2AX foci present 2 h following irradiation (8 Gy) in combined treatment conditions compared to irradiation alone. This influence on alterations in DNA damage resolution was also confirmed by a study investigating a different Chk1 inhibitor (SAR-020106) on the radiosensitisation of GBM models [96]. This study demonstrated increased sensitivity to radiation following pre-treatment with SAR-020106 and explored the persistence of SSBs via alkaline comet assays, revealing increased (1.7–2.1 fold) DNA damage in T98G and P0297 cell lines, when compared to irradiated controls. A further study, looking more specifically at DSBs via neutral comet assays, demonstrated an increase in the damage for up to 48 h post irradiation, highlighting persistent unrepairable DSBs following pre-treatment with the Chk1 inhibitor AZD7762 in UMSCC-1 head and neck squamous cell carcinoma (HNSCC) cells [97]. This was also demonstrated at an earlier time point of 4 h post irradiation when investigating γH2AX positive cells via flow cytometry analysis. Although this study demonstrated a significant increased radiosensitivity of p53 mutant UMSCC-1 cells with a dose enhancement ratio (DER) value of 2.94, this was not recapitulated in UMSCC-6 cells that have a functional p53 pathway. This indicates the p53-dependent radiosensitising potential of Chk1 inhibition. In support of this, a study investigating the Chk1 inhibitor MK-8776 in non-small cell lung cancer and HNSCC cell lines prior to irradiation observed increased radiosensitisation of p53-mutant cell lines (H1299, CaLu-6, FaDu) but not of p53-wild-type tumour cells (A549 and H460), with DER values of 1.25–1.53 and 1.0–1.03, respectively [98]. Analysis of γH2AX and 53BP1 foci revealed a significant increase in foci number and therefore of DSBs in H1299 lung cancer cells, but not A549 cells, at 30 min and 1 h post irradiation, respectively, following pre-treatment with MK-8776 compared to the irradiated controls.

The notion of p53 dependency was also seen in a study investigating the radiosensitisation of AZD7762 in isogenic colorectal (HCT116) and lung (H460) cell lines, where effects were only seen in p53-mutant strains [99]. Additionally, no radiosensitisation was observed in normal intestinal epithelial cells (CCL-241) harbouring functional p53, although cytotoxicity was evident. Moreover, a study utilising the SAR-020106 Chk1 inhibitor indicated that lung cancer cells (A549) with functional p53 were more likely to undergo G_1_/S arrest in order to repair the radiation-induced DNA damage and maintain genome stability promoting cellular survival [100]. In contrast to this evidence, a radiosensitisation effect was seen in both p53 mutant and functional isogenic colorectal (HCT116) cancer cell lines following CHIR-124 treatment [101]. Supporting this, Cal-51 breast cancer cells were effectively radiosensitised following MK-8776 pre-treatment, despite harbouring functional p53 [95]. Consequently, and from the data available, the influence of p53 status on the radiosensitising effect of Chk1 inhibition could be tumour type and/or inhibitor specific, although more investigative studies are required to fully determine whether p53 mutational status is indeed a pre-requisite for a preferential response to Chk1 inhibitors in combination with radiation.

**Table 1 cancers-16-03016-t001:** Radiosensitisation with Chk1 inhibitors using X-rays.

Inhibitor	Cells	Observations	Ref
AZD7762	UMSCC-1, UMSCC-6, UMSCC-47	UMSCC-1 (DER = 2.94) Increased γH2AX staining at 4 h Increased neutral comet tail moment at 48 h Increased apoptotic cell death	[97]
AZD7762	MiaPaCa-2	γH2AX foci persists up to 24 h Reduced RAD51 foci formation at 26–30 h Reduced HR in DR-GFP assay Increased sensitivity in vivo	[102]
AZD7762(+Olaparib)	MiaPaCa-2, MPanc-96, HCT116, H460, CCL-241	No radiosensitisation to p53-wt or epithelial cells AZD7762 (DER = 1.5–2.0), plus olaparib (DER = 2.4–3.0) Increased γH2AX foci at 16–24 h	[99]
AZD7762	MCF-7, A549, H460, HT29, 1522, PC-Sw, SF-295, DU145, MiaPaCa-2	p53-mut (DER = 1.6–1.7) p53-wt (DER = 1.1–1.2) Increased γH2AX expression at 8 and 24 h Increased nuclear fragmentation at 24–72 h Increased sensitivity in vivo	[103]
CCT244747	RT112, T24, Cal-27, hTertRPE1	DER = 1.33–1.62 Increase in pan-γH2AX and abnormal nuclei Increased cleaved PARP-1 and caspase-3 at 48 h Increased sensitivity in vivo with Cal-27 xenografts	[104]
CHIR-124	HCT116	Radiosensitisation irrespective of p53 status Radiosensitisation in p21, Chk2 and MAD2 KO cell lines Increased mitotic catastrophe in p53-mut	[101]
LY2606368	UMSCC-1, UMSCC-47	Increase in S phase cells Increased γH2AX, caspase-3 and apoptosis at 48 h Increased sensitivity in vivo xenografts	[105]
MK-8776	MDA-MB-231, BT-549, Cal-51	Radiosensitivity irrespective of p53 status Increased γH2AX staining at 2 h Cell death via autophagy Radiosensitivity in vivo	[95]
MK-8776	H1299, Calu-6, FaDu, A549, H460	No radiosensitisation in p53-proficient cells Increase in γH2AX/53BP1 foci and reduced RAD51 activity	[98]
MK-8776	MiaPaCa-2, BxPC-3, AsPC-1, Capan-1	Reduction in RAD51 foci formation	[106]
SAR-020106	LN405, T98G, A172, DBTRG, P0297, P0306	1.7–2.1 fold increase in SSBs at 24 h	[96]
SAR-020106	Cal-27, HeLa, HN6, A549, MEF	No radiosensitisation in p53-proficient cells Reduced γH2AX/RAD51 foci colocalization Unsuccessful cytokinesis and increase in aneuploid cells Increased apoptosis at 24 h Radiosensitivity in vivo, with increased TUNEL staining	[100]
UCN-01	HeLa	Prolonged mitosis due to SAC activation	[107]
UCN-01 MK-8776	HeLa	UCN-01 (DER = 1.07–1.13) MK-8776 (DER = 1.22–1.39) No alterations in γH2AX/RAD51 foci Increased centromere numbers and time spent in mitosis	[108]

It has been suggested that the increase in persistent DSB damage highlighted in multiple studies post irradiation following pre-treatment with a variety of Chk1 inhibitors, could be linked to a reduced HR proficiency of the cells. It is known that Chk1 has a direct action on RAD51 through phosphorylation of the threonine 309 site, which influences the stability and regulation of RAD51, with some reports suggesting an influence on interactions with BRCA2 and sub-cellular localisation to the damage [36,109]. In addition to direct effects on RAD51, it has been suggested that the limited HR response could be due to abrogation of the radiation-induced G_2_/M arrest and subsequently cells with damaged DNA no longer reside in the appropriate cell cycle phases for effective HR to occur. The direct influence of Chk1 inhibition via AZD7762 on RAD51 activity was investigated in MiaPaCa-2 pancreatic cancer cells, where it was observed that RAD51 foci formation were prevented at 26 and 30 h post-irradiation, whereas there was an increase in γH2AX foci 24 h following the combination treatment [102]. These results indicated that Chk1 inhibition directly affected the formation but not disassociation of RAD51 foci, which was further supported with the DR-GFP assay showing reduced HR activity on DSBs following AZD7762 treatment. In a separate study, the potential of combining AZD7762 with the PARP-1 inhibitor Olaparib to enhance radiosensitivity was explored, given that BRCA-ness is known to be synthetically lethal with PARP-1 inhibition [110,111]. This study indicated enhanced radiosensitivity of both MiaPaCa-2 and MPanc-96 pancreatic cancer cells following pre-treatment of AZD7762 with DER values of 1.5–2.0 [99]. Interestingly, cellular radiosensitivity to Olaparib (DER value of 1.5) was significantly increased following the addition of AZD7762 (DER values of 2.4–3.0). Furthermore, this study showed an increase in γH2AX-positive cells at both 16 and 24 h following the pre-treatment of both AZD7762 and Olaparib, compared to any other treatment conditions, indicating significant increases in the level of unrepaired DSBs likely through a reduced HR response. Moreover, another study in pancreatic cancer models demonstrated an increase in radiosensitivity and a reduction in RAD51 foci formation following MK-8776 treatment with X-ray radiation and gemcitabine in MiaPaCa-2, BxPC-3 and AsPC-1 cell lines, however, this effect was not evident in BRCA-2-deficient Capan-1 cell lines [106]. Similarly, pre-treatment with the Chk1 inhibitor SAR-020106 revealed increased radiosensitivity in cells harbouring p53-mutations (Cal-27, HeLa and HN6) with associated decreases in γH2AX/RAD51 foci co-localisation, indicating reduced HR efficiency [100].

In addition to influencing the effectiveness of HR, Chk1 inhibition has been shown to cause irradiated cells to prematurely enter mitosis. A study in bladder cancer and HNSCC cells (RT112 and Cal-27) demonstrated a significant increase in the mitotic population following the addition of CCT24474 6 h post irradiation, associated with enhanced cellular radiosensitisation (DER values of 1.33–1.62) [104]. This early entry into mitosis with unresolved DNA damage has been shown to increase abnormal nuclei formation and nuclear fragmentation, associated with mitotic catastrophe. Indeed, nuclear fragmentation was observed to increase 24–72 h post irradiation following pre-treatment with the Chk1 inhibitor AZD7762 in p53-mutant H460 lung cancer cells [103], further supported by observations of a similar increase in nuclear fragmentation at 24 h following CEP-3891 pre-treatment in U2OS cell lines, shown to be as a result of abnormal chromosomal segregation [112]. Additionally, a study in isogenic HCT116 colorectal cancer cell lines observed increases in micronuclear and multinuclear cells post-irradiation following pre-treatment with the Chk1 inhibitor CHIR-124 [101]. Interestingly, this study also observed an increase in the radiosensitivity of MAD2-deficient HCT116 cell lines (DER value of 1.70), an essential protein in the SAC. However, significant increases in SAC activation were observed in HeLa cells, along with increases in mitotic catastrophe and subsequent cell death following pre-treatment with UCN-01 and a high 15 Gy dose of radiation [107]. Additionally, a study comparing the effects of UCN-01 and MK-8776 following irradiation in HeLa cells found that although irradiation alone increased time spent in mitosis, this was significantly further increased by Chk1 inhibition [108]. It was determined that UCN-01 treated cells resulted in 40% of cells in aberrant mitosis, whereas the effects of MK-8776 were much more profound with 75% of cells being affected, likely due to the increased specificity of MK-8776. Premature mitotic entry and subsequent nuclear fragmentation has also been shown to result in apoptosis, by increased caspase activity, PARP-1 cleavage and annexin-V analysis 48 h following irradiation in a number of different cancer cell lines (RT112, UMSCC-1, UMSCC-47 and U2OS) [104,105,112], which was further supported by immunohistochemical analysis of TUNEL staining in Cal-27 xenograft tumours [100]. Furthermore, multiple studies have shown significant radiosensitisation of tumours in vivo, both in xenograft and patient-derived xenograft (PDX) mouse models, utilising various Chk1 inhibitors (AZD7762, MK-8776, CCT244747, LY2606368, SAR-020106) [95,100,102,103,104,105]. These studies overall have demonstrated significant reductions in tumour growth and an increase in overall survival of the mice compared to irradiation alone, and which have supported clinical progression.

Overall, the consensus of pre-clinical evidence suggests that inhibiting cell cycle checkpoint activation post-irradiation via Chk1 inhibition results in an increase in unrepairable DNA damage, pre-mature mitotic entry, chromosomal aberrations and mitotic catastrophe, subsequently increasing tumour cell death.

### 4.4. Wee1 Inhibitor Preclinical Studies

Pre-clinical evidence for the use of Wee1 inhibition as a potential radiosensitiser is summarised (Table 2). Inhibiting the Wee1 kinase as an approach for radiosensitising tumour cells was proposed following the development of the first Wee1 inhibitor PD0166285 [85]. This study observed the sensitising potential of inhibiting Wee1 in a variety of tumour cell types in a p53-null dependent manner as exhibited by DER values of 1.23–1.38. This was further confirmed by a follow-up study utilising temperature sensitive inducible p53 expressing H1299 lung carcinoma cells, again indicating that enhanced radiosensitivity was dependent on the absence of p53 (DER values of 1.25 and 1.09 in p53-deficient and proficient cell lines, respectively) [113]. Later work with the more selective Wee1 inhibitor MK-1775 in a variety of tumour types has also supported this, identifying DER values of 1.2–1.5 for p53-mutant cell lines, with no observable radiosensitivity in p53-containing cells [114]. Interestingly, there is conflicting evidence to suggest that radiosensitisation with Wee1 inhibition (MK-1775) is irrespective of the cells p53 status, observed in both GBM and HNSCC cell models [115,116].

In terms of mechanistic evidence supporting a role for DNA damage in the radiosensitisation of cells with Wee1 inhibition, a study in osteosarcoma cell models (MG63, U2OS and SaOS-2) revealed increased DSB damage via persistent γH2AX foci 24 h post irradiation [125], which was further supported by work in VUMC-DIPG-A and E98 glioma cell lines indicating persistence of γH2AX and 53BP1 foci for up 72 h post irradiation [117]. Additionally, a reduction in γH2AX foci resolution was observed following the combination of MK-1775 plus radiation compared to radiation alone in MiaPaCa-2 and MPanc-1 pancreatic cancer cells, but not in Capan-1 (BRCA-2-deficient) or HR-modified DLD-1 cells, which corresponded with the observed radiosensitivity [118]. A reduced HR response following Wee1 inhibition was also seen in KYSE150 and TE1 oesophageal cancer cell lines treated with PD0166285, as observed by increases in γH2AX foci and alkaline comet tail DNA indicative of SSBs 24 h post irradiation along with a decrease in RAD51 foci formation at 12 and 24 h following the combination treatment, correlating with enhanced radiosensitisation (DER values of 1.37–1.60) [126]. It has also been confirmed that Wee1 inhibition via either MK-1775 or PD0166285 increases γH2AX foci but reduces formation of 53BP1 foci and HR proficiency in both MCF-7 and HeLa cells via the DR-GFP reporter assay [124]. This coincides with the synthetic lethality recently demonstrated with MK-1775 and PARP inhibition [128,129], the combination of which can increase sensitivity of cells to radiation [120,130]. Interestingly, a study in HepG2, Hep3B and Huh7 cell lines identified increased pan-nuclear γH2AX staining, an indicator of replication stress, 16 h post-irradiation following treatment with MK-1775 [119]. This study also demonstrated that nucleotide supplementation successfully reversed the enhanced radiosensitivity of Huh7 cells following MK-1775 treatment, with DER values decreasing from 1.22 to 0.96. This is further supported by another study demonstrating the ability to overcome the radiosensitising phenotype of MK-1775 through nucleotide supplementation in Calu-6, H23 and H1730 cell lines [120]. However, this was only a partial rescue, as the percentage of pan-nuclear γH2AX cells reduced (from ~35–40% to ~15%) following nucleotide supplementation in combinatorial treatment conditions. This relatively high level of replication stress (~15%) was also evident following single-agent treatment with MK-1775 and which is supported by a number of other studies demonstrating that Chk1 and Wee1 inhibitors alone can cause replication stress [44,131,132,133,134].

In addition to causing genome instability, it has been shown that Wee1 inhibition results in pre-mature mitotic entry as early as 4 h post irradiation in H1299 lung carcinoma cells [114]. This was also observed in HepG2, Hep3B, Huh7, Cal-6, H23 and H1730 cell lines via analysis of phosphorylated histone H3 [119,120]. Furthermore, it was evident that these early mitotic cells were accompanied with increases in micronuclei formation and nuclear fragmentation, indicative of mitotic catastrophe [116,127]. However, a study in HNSCC cell lines suggested that p53 status is an important determinant in the mechanism of cell death triggered by MK-1775, whereby p53-containing cell lines (UMSCC-47) result in mitotic catastrophe, whereas, p53-mutant cells (SCC-15, SCC-25 and Cal-27) increase apoptosis following combination treatment [115]. However, it should be noted that this shift in cell death dependency could be due to human papillomavirus status (which UMSCC-47 cells contain) and not p53, which could prevent the formation of apoptotic bodies but which nevertheless requires further exploration. In contrast to this theory, an increase in apoptosis has been seen in HeLa and siHa cell lines via analysis of Annexin V staining 72 h post irradiation and an increase in caspase activity at 48 h [121], further supported by an increase in the same end-points in MG-63, U2OS and SaOS-2 cell lines [125].

The radiosensitising potential of Wee1 inhibition, in addition to cell lines, has also been shown in xenograft and PDX in vivo models through decreases in tumour growth and increased overall survival of the mice [114,116,120,121,122,126,127]. These studies have explored the influences of both PD0166285 and MK-1775 as Wee1 inhibitors showing no observable toxicities, which correlated with the lack of radiosensitivity phenotype seen in normal human cell lines when investigated in vitro [116,123,125]. Despite this, it is noteworthy that a study utilising MK-1775 in GBM stem cell populations (G179 and G144) post-irradiation indicated an abrogation of the G_2_/M arrest, although there were no influences on radiosensitivity levels, unlike the radiosensitisation observed in non-stem cell-like GBM cell lines (U251, U87 and T98G) [116]. This contrasts with work demonstrating that the combination of radiation with the broadly active Wee1 inhibitor PD0166285 in stem-like GBM primary neurosphere models did achieve significantly enhanced radiosensitisation [127]. This nevertheless highlights that the ability of Wee1 inhibition to overcome the inherent radioresistance of cancer stem cell populations requires further elucidation. Collectively, pre-clinical evidence for targeting Wee1 in combination with radiation suggests this as a promising therapeutic avenue in a number of different tumour types.

### 4.5. Chk1/Wee1 Inhibitors in Combination with Protons and Higher LET Radiation

Studies examining the combination of Chk1 or Wee1 inhibitors with alternative radiotherapies (particularly those of a higher LET, which are therefore more densely ionising than low-LET X-rays) are distinctly lacking (Table 3). A study of TNBC cells has revealed that G_2_/M checkpoint arrest was induced to a greater degree following 230 MeV protons compared to X-ray irradiation, and subsequently, an enhanced radiosensitising effect of a Chk1 inhibitor (PF-00477736) following protons was observed [135]. This effect is likely proton-specific, rather than being related to LET (and therefore to DNA damage complexity) given the low-LET nature of the proton irradiation used. However, in contrast, a study performed in 3D pancreatic cancer models (Colo357 and MiaPaCa-2) found an enhanced radiosensitising effect following X-rays (DER values of 1.5–1.6), compared to low-LET proton irradiation (DER values of 1.3) following Chk1 inhibition via LY2606368 [136]. A study targeting Wee1 (MK-1775) in non-small cell lung cancer cells exposed to X-rays or carbon ions (50 keV/µm) revealed no significant difference in the degree of radiosensitisation comparing both modalities, therefore suggesting no relation of this effect to LET [137]. Similarly, another study conducted in non-small cell lung cancer cells showed that a Chk1/Chk2 inhibitor (AZD7762) radiosensitised the cells to X-ray irradiation as well as to carbon ions (50 keV/µm) to a similar degree in one cell line (A549), although the response to carbon ions was exacerbated in another (H1299) [138]. These effects were considered to be driven through a more profound abrogation of the G_2_/M checkpoint arrest and more persistent and complex DSBs following carbon ion irradiation. This study therefore indicates that the effects of cell cycle inhibition could be more effective following high-LET radiation, which is further supported by a study performed in HNSCC stem-like cancer cells [139]. Here, radiosensitisation following X-ray irradiation in cancer stem-like cells following Chk1 inhibition (UCN-01) was observed, although the degree of radiosensitisation was much more pronounced following high-LET carbon ions (184 keV/µm). Nevertheless, substantially more studies using PBT and high-LET particle ions are required to further understand the therapeutic potential of targeting cell cycle checkpoint proteins to increase the therapeutic efficacy of the radiotherapy modalities, and to identify any associated relationship to LET.

### 4.6. Clinical Trials of Chk1/Wee1 Inhibitors in Combination with Radiotherapy

There has only been one clinical trial to date examining the effect of Chk1 inhibition (LY2606368) in combination with radiotherapy, which was centred on HNSCC patients. However, significantly more trials have been conducted with the Wee1 inhibitor (MK1775/AZD1775) in different cancer types (Table 4). Unfortunately, the combination of AZD1775 with cisplatin and radiotherapy has shown toxicities in patients with head and neck cancers [140]. However, some promising results were shown with Wee1 inhibition in combination with gemcitabine and radiotherapy for pancreatic cancer patients [141]. Interestingly, Wee1 inhibition in clinical trials as a single agent therapy has shown preferential responsiveness in uterine and ovarian tumours, specifically harbouring BRCA mutations [142,143,144,145]; however, only one study investigating radiosensitive effects in these cancer types has been explored and was terminated early due to limiting toxicities (NCT03345784). Nevertheless, based on clinical trials to date, it is apparent that a precise combination of treatments utilising Chk1 or Wee1 inhibitors and the specific disease setting need to be further explored. In an effort to better stratify patients, multiple clinical trials have recently completed or are still ongoing to identify clinical biomarkers for increased responsiveness to both Chk1 and Wee1 inhibitors as monotherapies, which could later be translated into effectively selecting patients for treatment in combination with radiotherapy (Chk1; NCT02873975, NCT02203513, NCT05548296, NCT02797964, Wee1; NCT03668340, NCT03385655, NCT03253679, NCT01748825, NCT04439227, NCT02482311). Some potential biomarkers highlighted include BRCA mutations, CCNE1 or MYC amplification.

## 5. Conclusions and Future Directions

Chk1 and Wee1 are important protein kinases that control the G_2_/M cell cycle checkpoint, which is triggered in cells to allow DNA damage repair before entering mitosis. Consequently, targeting Chk1 and Wee1 with specific inhibitors in combination with radiotherapy to induce DNA damage is considered a potentially important therapeutic strategy for treating human cancers, particularly those with p53 mutations that lack an efficient G_1_/S checkpoint and therefore are predictably more susceptible to this treatment. Whilst the first Chk1 inhibitor was developed over two decades ago, more specific and potent inhibitors have since been generated. Wee1 is a downstream target for Chk1 kinase activity, and whilst similar to Chk1, this is considered an interesting cancer therapeutic target, although the development of more specific inhibitors of Wee1 has been lacking. Nevertheless, there is accumulating preclinical evidence to demonstrate that both Chk1 and Wee1 inhibition are able to radiosensitise cells from various different tumours both in vitro and in vivo, which supports their potential to be exploited further in the clinic for the benefit of cancer patients.

Whilst the potential for Chk1 and Wee1 inhibitors to act as tumour radiosensitisers is compelling, there remain some uncertainties, and the mechanisms ultimately leading to cell death in the defined tumour cell models needs to be further investigated (Figure 3). A significant number of studies indicate persistent radiation-induced DNA damage caused by the lack of a G_2_/M checkpoint through Chk1/Wee1 inhibition, which ultimately leads to mitotic catastrophe. Equally, though, this also could be created through direct suppression of HR repair of DSBs. There is furthermore evidence to suggest that Chk1/Wee1 inhibitors with radiation lead to replication stress, as well as triggering alternative mechanisms of cell death (apoptosis and autophagy), which need to be explored further to understand the molecular mechanisms through which these are driven. Interestingly, there is some evidence to suggest that tumour cells containing functional and wild-type p53 can be radiosensitised following Chk1 and particularly Wee1 inhibition, which would be unexpected given that these cells have an active G_1_/S cell cycle checkpoint that should enable efficient DNA damage repair. This could also have implications for tumour-associated normal cells containing functionally active p53, wherein biological evidence of the impact of Chk1 and Wee1 inhibitors in combination with radiotherapy is lacking. Indeed, there is some evidence to suggest that Wee1 inhibition, albeit in combination with both radiation and cisplatin, produces dose-limiting toxicities in patients with HNSCC, which could in part be explained by our lack of knowledge of the normal tissue responses to this combinatorial treatment. Going forward, more efforts need to be made to understand the influence of p53 on responses to Chk1/Wee1 inhibitors with radiation in both tumour and normal tissues, with a view to decreasing any potential adverse side effects of the treatment and increasing the therapeutic ratio.

Related to the reduction of radiotherapy-induced side effects is the utilisation of more precision-targeted PBT that, compared to photon radiotherapy, will reduce the amount of radiation dose delivered to the normal tissues and organs at risk. However, again, there is minimal biological evidence exploring the potential of combining Chk1 or Wee1 inhibitors with alternative radiation modalities, such as protons and high-LET carbon ions. Interestingly, there is some evidence to suggest that the radiotherapy type or LET (and therefore the amount of CDD) could influence the biological response to Chk1/Wee1 inhibition, possibly related to the degree or strength of activation of the G_2_/M checkpoint. More systematic studies comparing the impact of different radiation sources in well-defined tumour (and normal) models are therefore needed to understand the potential for Chk1 or Wee1 inhibitors to be utilised in order to optimise tumour radiosensitisation to the various radiation modalities. Finally, it is still unclear which specific tumours will significantly benefit from Chk1/Wee1 inhibition in combination with radiotherapy in the clinic. In this review, we have highlighted numerous preclinical studies that have been performed in various tumour cell types, but also clinical trials largely centred on HNSCC and GBM. Apart from p53 status, there needs to be more investigation and identification of the biomarkers of treatment response to understand and stratify patients that will benefit from Chk1/Wee1 inhibition with radiotherapy. For now, Chk1 and Wee1 remain important cancer targets for radiosensitisation, but they need further experimental exploration.

## Figures and Tables

**Figure 1 cancers-16-03016-f001:**
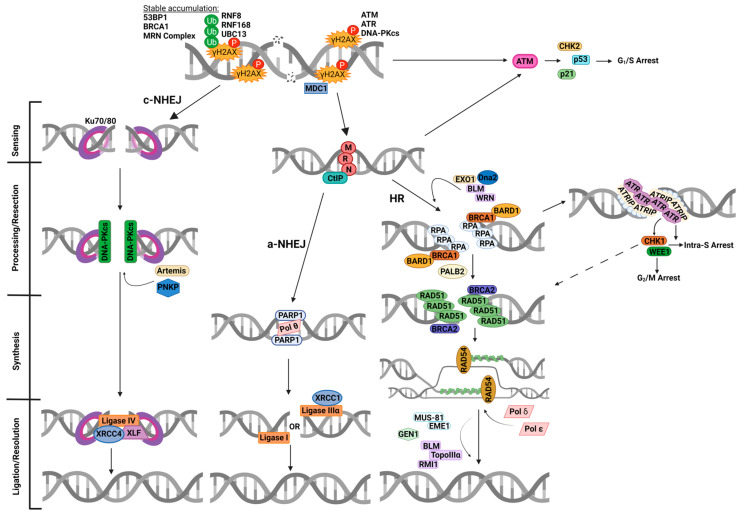
Overview of the cellular DDR to DSBs. Ionising radiation induces DSBs, leading to γH2AX formation stimulated by ATM, ATR and DNA-Pkcs, which recruits multiple DNA damage response proteins, including MDC1. γH2AX is targeted for ubiquitination by the RNF8-RNF168-UBC13 complex, enabling accumulation of further DDR proteins, including 53BP1, BRCA1 and the MRN complex. DSB breaks can be repaired by either NHEJ (c-NHEJ or a-NHEJ) or HR. In c-NHEJ, Ku70/80 heterodimers bind to and anchor the damaged DNA ends, which recruit DNA-Pkcs and other end-processing proteins, such as Artemis and PNKP, before ligation occurs via the XRCC4-Ligase IV-XLF complex. Alternatively in a-NHEJ, the MRN complex can bind to the DNA ends and initiate end resection with CtIP. Following this, PARP-1 binds to the DNA ends, allowing for synthesis within the break by Pol θ and ligation by either Ligase I or XRCC1-Ligase IIIα. Binding of the MRN-complex also recruits and activates ATM, which allows for a signalling cascade through Chk2, p53 and p21 for arrest at the G_1_/S checkpoint. During HR, EXO1, Dna2, BLM and WRN can be associated with end resection in addition to the MRN complex and CtIP, which results in ssDNA that is then coated by RPA. This simulates the activation of ATR via ATRIP which results in intra-S and G_2_/M checkpoint arrest, orchestrated via Chk1 and Wee1 activity. Following this, BRCA1 and its dimerization partner BARD1 interact with PALB2 and recruit BRCA2 and RAD51. RAD51 then replaces RPA to form nucleofilaments on ssDNA. The activity of RAD51 is also influenced by Chk1 (dashed arrow). RAD51-ssDNA then undergoes homology search and invasion of the sister chromatid, facilitated by BRCA2 and RAD54, and DNA synthesis is completed by either Pol δ or Pol ε. DNA ligation then forms Holliday junctions which are processed by resolvases, including MUS81-EME1/2, GEN1 and BLM-TopoIIIα-RMI1. This figure was created with Biorender.com.

**Figure 2 cancers-16-03016-f002:**
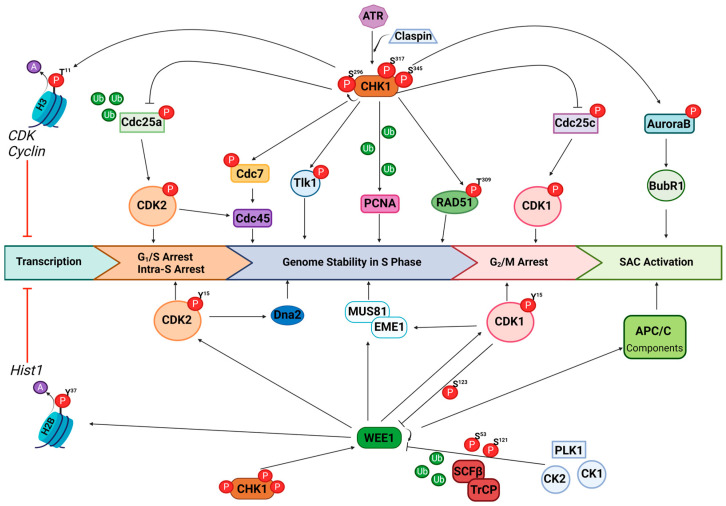
The major signalling roles of Chk1 and Wee1. Chk1 is phosphorylated by activated ATR at serine 317 and 345, via interaction with Claspin adaptor protein, which stimulates autophosphorylation of Chk1 at serine 296. Activated Chk1 phosphorylates histone H3 at threonine 11, which reduces acetylation and ultimately reduces transcription of various CDK and cyclin genes. The phosphatase Cdc25a is directly phosphorylated by Chk1 which targets this for ubiquitination and reduces dephosphorylation of CDK2, resulting in arrest at intra-S and G_1_/S checkpoints. CDK2 also influences the loading of Cdc45 during DNA replication, which is further controlled by Cdc7 that is directly phosphorylated by Chk1. Additionally, Chk1 influences genome stability during replication via phosphorylation of Tlk1 and targeting PCNA for ubiquitination. RAD51 is directly phosphorylated by Chk1 at threonine 309, which influences both DNA replication and HR. Direct phosphorylation of the phosphatase Cdc25c prevents dephosphorylation of CDK1, which results in G_2_/M arrest. Finally, Chk1 influences SAC activation through phosphorylation of Aurora B, which impacts the localisation of Bub1. Wee1 is also a direct target for activation by Chk1. Activated Wee1 causes phosphorylation of histone H2B at tyrosine 37, which reduces acetylation and prevents the transcription of the *Hist1* gene cluster. Wee1 also directly phosphorylates CDK2 and CDK1 at tyrosine 15, resulting in intra-S-G_1_/S and G_2_/M checkpoint arrest, respectively. Phosphorylated CDK1, furthermore, is able to influence MUS81-EME1 required for genome stability. Wee1 also targets various APC/C components for phosphorylation and therefore influences the activation of SAC. Finally, CDK1 stimulated phosphorylation of Wee1 at serine 123 allows for further phosphorylation via PLK1 and CK1/2 at serines 53 and 121, thus targeting Wee1 for degradation via the SCFβ-TrCP complex.

**Figure 3 cancers-16-03016-f003:**
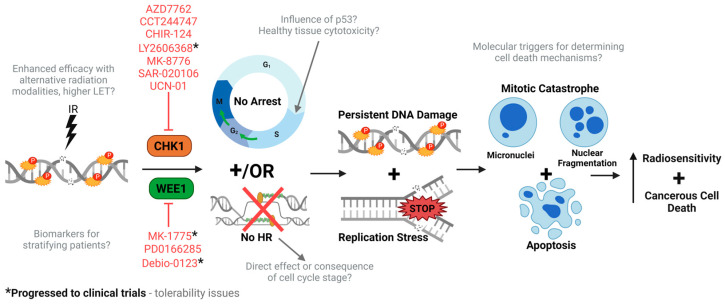
Overview of our current understanding of Chk1 and Wee1 inhibitors as radiosensitisers. Ionising radiation induces DNA damage, specifically to DSBs, which stimulates the cellular DDR including cell cycle arrest. In combination with Chk1 or Wee1 inhibitors, radiation-induced arrest at both intra-S and G_2_/M checkpoint is abrogated, and HR repair is limited. This forces the cells through the cell cycle under replication stress and harbouring unrepaired DNA damage, resulting in mitotic catastrophe (micronuclei/fragmentation) and apoptosis, ultimately increasing cellular radiosensitivity. Preclinical evidence demonstrating the radiosensitisation potential of Chk1 and Wee1 inhibitors is listed in red, with those progressing to clinical trials (indicated with *). Despite this, there are still some uncertainties regarding the underlying mechanisms of action (highlighted in grey).

**Table 2 cancers-16-03016-t002:** Radiosensitisation with Wee1 inhibitors using X-rays.

Inhibitor	Cells	Observations	Ref
MK-1775	A549, H1299, Calu-6, H460, CCD16, MCF-7, MDA-MB-231, MCF-10A, PC3, LnCaP	No radiosensitivity in p53-proficient cells Radiosensitivity in p53-deficient cells (DER = 1.2–1.5) Increased γH2AX Pre-mature mitotic entry at 4 h Increased micronuclei Radiosensitivity in vivo	[114]
MK-1775	UMSCC-47, SSC-25, SSC-15, Cal-27	Radiosensitisation irrespective of p53 status p53-containing cells increased mitotic catastrophe p53-deficient cells increased apoptosis	[115]
MK-1775	U251, U87, T98G, G179, G144, astrocytes	Radiosensitisation irrespective of p53 status (DER = 1.2–1.3) No radiosensitivity of stem cells Slight radioprotective in astrocytes Increase γH2AX expression at 10 h Increase in micronuclei formation Radiosensitivity in vivo	[116]
MK-1775	VUMC-DIPG-A, E98	γH2AX and 53BP1 foci up to 72 h	[117]
MK-1775	MiaPaCa-2, Panc-1, Capan-1, DLD-1	Increased γH2AX foci No radiosensitisation in HR-deficient cells Radiosensitivity in vivo, decreased RAD51 foci formation	[118]
MK-1775	HepG2, Hep3B, Huh7	Radiosensitivity (DER = 1.22–1.38) Increased pan-nuclear γH2AX at 16 h Nucleotide supplementation reversed radiosensitivity Pre-mature mitotic entry	[119]
MK-1775	Calu-6, H23, H1730	Radiosensitivity (DER = 1.23–1.43) Nucleotide supplementation reversed radiosensitivity Pre-mature mitotic entry Radiosensitivity in vivo	[120]
MK-1775	HeLa, siHa	Increased γH2AX expression Increased caspase activity and apoptosis at 48–72 h Radiosensitivity in vivo; increased yH2AX/TUNEL, decreased Ki67	[121]
MK-1775	OE33, SK4, KYSE30, AGS	Radiosensitivity (DER = 1.23–3.14) Increased γH2AX expression at 24 h Increased mitotic catastrophe Radiosensitivity in vivo; increased mitotic catastrophe	[122]
MK-1775	Jurkat, MOLT-4	No radiosensitivity in normal cells Increased γH2AX staining	[123]
MK-1775 PD0166285	BJ fibroblasts, murine cardiomyocytes, MCF-7, HeLa, MDA-MB-231, SK-BR-3, T47D	Radiosensitisation in p53-deficient cells Increased γH2AX foci Reduced 53BP1 foci Reduced HR proficiency via DR-GFP	[124]
PD0166285	PA-1, HT29, HeLa, HCT8, HCT116, DLD-1, H460, C26	Radiosensitisation in p53-deficient cells (DER = 1.23–1.38) Increased mitotic index	[85]
PD0166285	H1299, MCF-7	Radiosensitisation in p53-deficient H1299 cells (DER = 1.25) but not in p53-expressing H1299 cells (DER = 1.09)	[113]
PD0166285	MG-63, U2OS, SaOS-2, Hum31, Hum54	No radiosensitisation in normal cells Increased γH2AX foci at 24 h Increased caspase activity and apoptosis	[125]
PD0166285	KYSE70, KYSE150, KYSE410, KYSE450, KYSE510, TE1, TE7, EC1, HEEC	Radiosensitivity (DER = 1.37–1.60) Increased γH2AX foci and SSBs at 24 h Decreased RAD51 foci formation Increased mitotic catastrophe, caspase activity and apoptosis Radiosensitivity in vivo; decreased RAD51	[126]
PD0166285	U251-MG, U118-MG, U87-MG, U373-MG, VU147, VU148, E98, fibroblasts, astrocytes	Radiosensitivity (DER = 1.19–1.95) Increased radiosensitivity of primary stem-like cells Increase γH2AX foci Increase in pre-mature mitotic entry and nuclear fragmentation Radiosensitivity in vivo	[127]

**Table 3 cancers-16-03016-t003:** Radiosensitisation with Chk1 and Wee1 inhibitors using high-LET radiation.

Target	Inhibitor	Cells	Radiotherapy Type	Observations	Ref
Chk1	PF-00477736	MDA-MB-231, Hs578T	280 MeV protons	Enhanced radiosensitivity of protons versus X-rays	[135]
Chk1	UCN-01	Stem-like subpopulation of SQ20B	11.4 MeV carbon ions (184 keV/µm)	Enhanced radiosensitisation with carbon ions versus X-rays	[139]
Chk1/Chk2	AZD7762	A549, H1299	80 MeV carbon ions (50 keV/µm)	Equal radiosensitisation with carbon ions and X-rays in A549, but enhanced radiosensitisation with carbon ions in H1299	[138]
Chk1/Chk2	LY2606368	Colo357, MiaPaCa-2	150 MeV protons (3.7 keV/µm)	Enhanced radiosensitivity of X-rays versus protons	[136]
Wee1	MK1775	H1299	290 MeV carbon ions (50 keV/µm)	Equal radiosensitisation with carbon ions and X-rays	[137]

**Table 4 cancers-16-03016-t004:** Clinical trials with Chk1 and Wee1 inhibitors in combination with radiotherapy.

Target	Identifier	Phase	Inhibitor/Treatment	Tumour	Status/Outcomes	Ref
Chk1/Chk2	NCT02555644	Ib	LY2606368 plus cisplatin/cetuximab and RT	HNSCC	Completed in 2019	
Wee1	NCT03345784	I	AZD1775 plus cisplatin and RT	Cervical, upper viaginal, uterine	Closed prematurely; limiting toxicities	
Wee1	NCT01922076	I	AZD1775 and RT	DIPG	Completed in 2022	
Wee1	NCT05765812	I/II	Debio 0123 plus TMZ and RT	GBM	Active/recruiting	
Wee1	NCT01849146	I	AZD1775 plus TMZ and RT	GBM	Active/not recruiting	
Wee1	NCT03028766	I	AZD1775 plus cisplatin and RT	HNSCC	Completed in 2021; poor recruitment and toxicities	
Wee1	NCT02585973	Ib	AZD1775 plus cisplatin and RT	HNSCC	Completed in 2021; dose-limiting toxicities	[140]
Wee1	NCT04460937	I	AZD1775 and RT	Oesophageal and gastrooesophageal	Active/not recruiting	
Wee1	NCT02037230	I	MK1775 plus gemcitabine and RT	PDAC	Completed in 2018; treatment well tolerated and improved overall survival	[141]

Abbreviations: DIPG, diffuse intrinsic pontine glioma; GBM, glioblastoma; HNSCC, head and neck squamous cell carcinoma; PDAC, pancreatic ductal adenocarcinoma; RT, radiotherapy; TMZ, Temozolomide.

## Data Availability

No new data were created in this review. Therefore, data sharing is not applicable.

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
