# Peer review of "The Potential for Targeting G2/M Cell Cycle Checkpoint Kinases in Enhancing the Efficacy of Radiotherapy"

_cancers, 2024, doi:10.3390/cancers16173016_

Round 1
Reviewer 1 Report
Comments and Suggestions for Authors
The review by E. Melia and J. Parsons covers the topic of G2 checkpoint abrogation by CHK1 and WEE1 inhibitors for tumor radiosensitization. The hypothesis that abrogation of the radiation-induced G2 checkpoint would selectively target p53-negative cells is actually quit old (>20 years). Many previous reviews have covered this topic in the past. Nevertheless, this review offers valuable new insights, particularly on the recent advancements of WEE1 inhibitors. The discussion regarding high-LET irradiation and CHK1/WEE1 inhibitors also contains new information, although this topic has also recently been reviewed by others. The review is clearly written, but a few issues are unclear or could have been discussed more thoroughly.
Specific comments:
1. Figure 1: From this figure, it looks like HR is upstream of ATR activation. However, DNA end resection is upstream of ATR activation, not HR repair? This should be corrected. Furthermore, from this figure, it looks like MRN and CtIP are upstream of radiation-induced G1/S arrest. Could the authors provide references to original articles showing that the radiation-induced G1/S depends on MRN and CtIP? (In the case such references do not exist, the figure should be corrected accordingly.)
2. Line 175-176: references are lacking.
3. Figure 2: It is not clear whether "Genome Stability" refers to cells in S-phase. Since this box is placed after "G1/S arrest/Intra-S arrest" and before "G2/M arrest/SAC activation", it most likely refers to Genome Stability in S-phase? Furthermore, it has previously been shown that CDK1/CDK2 suppression by WEE1 inhibition protects genome stability through control of replication initiation and nucleotide consumption. This information is lacking from the figure.
4. It is briefly mentioned that WEE1 inhibition can radiosensitize cells irrespective of p53 status. However, a discussion of whether such conflicting results have also been observed for CHK1 inhibition is lacking. It may be useful to discuss previous studies of isogenic cell systems only differing in p53-status, as well as studies examining panels of cancer cells cell lines with different p53 status. If such inhibitors can radiosensitize cells irrespective of p53-status, would they also radiosensitize normal cells?
5. Previous preclinical evaluation of WEE1 or CHK1 inhibition as single-agent anticancer therapy (without irradiation) has shown large variation between different cancer cell lines. The differences in single-agent activity would be relevant also for the combined effects with irradiation, but have not been discussed.
6. In line 582 it is mentioned that CHK1/WEE1 inhibitors with radiation lead to replication stress. Is this a specific effect for the combination with radiation? This issue could have been discussed more thoroughly. Many previous studies have shown that such inhibitors cause replication stress even in the absence of radiation. For instance, CHK1 or WEE1 inhibition causes aberrant replication initiation and subsequent DNA breakage in S-phase. Additionally, CHK1 and WEE1 inhibition may also protect against replication fork collapse.
Author Response
Reviewer 1 Comments and Suggestions for Authors
The review by E. Melia and J. Parsons covers the topic of G2 checkpoint abrogation by CHK1 and WEE1 inhibitors for tumor radiosensitization. The hypothesis that abrogation of the radiation-induced G2 checkpoint would selectively target p53-negative cells is actually quit old (>20 years). Many previous reviews have covered this topic in the past. Nevertheless, this review offers valuable new insights, particularly on the recent advancements of WEE1 inhibitors. The discussion regarding high-LET irradiation and CHK1/WEE1 inhibitors also contains new information, although this topic has also recently been reviewed by others. The review is clearly written, but a few issues are unclear or could have been discussed more thoroughly.
Specific comments:
- Figure 1: From this figure, it looks like HR is upstream of ATR activation. However, DNA end resection is upstream of ATR activation, not HR repair? This should be corrected. Furthermore, from this figure, it looks like MRN and CtIP are upstream of radiation-induced G1/S arrest. Could the authors provide references to original articles showing that the radiation-induced G1/S depends on MRN and CtIP? (In the case such references do not exist, the figure should be corrected accordingly.)
Response: We thank the Reviewer for their comments on Figure 1. We have now revised the figure to demonstrate more clearly that G1/S arrest is driven through ATM-Chk2 dependent pathway signalling of the DSB, whereas G2/M arrest is via ATR-Chk1 triggered through ssDNA coated by RPA following DNA end resection. The appropriate text has also been modified accordingly (lines 82-89).
- Line 175-176: references are lacking.
Response: We appreciate the Reviewer for this comment, and the references have now been added (lines 180-181).
- Figure 2: It is not clear whether "Genome Stability" refers to cells in S-phase. Since this box is placed after "G1/S arrest/Intra-S arrest" and before "G2/M arrest/SAC activation", it most likely refers to Genome Stability in S-phase? Furthermore, it has previously been shown that CDK1/CDK2 suppression by WEE1 inhibition protects genome stability through control of replication initiation and nucleotide consumption. This information is lacking from the figure.
Response: We thank the Reviewer for their comment. Figure 2 has now been altered to reflect the maintenance of genome stability specifically in S phase. We have also added into the text (lines 236-241) about the suppression of CDK2 activity by Wee1 that limits fork degradation by Dna2, controls replication origin firing and reduces nucleotide depletion.
- It is briefly mentioned that WEE1 inhibition can radiosensitize cells irrespective of p53 status. However, a discussion of whether such conflicting results have also been observed for CHK1 inhibition is lacking. It may be useful to discuss previous studies of isogenic cell systems only differing in p53-status, as well as studies examining panels of cancer cells cell lines with different p53 status. If such inhibitors can radiosensitize cells irrespective of p53-status, would they also radiosensitize normal cells?
Response: We thank the Reviewer for this important comment. As suggested, we have now added further details discussing the impact of p53 status on radiosensitisation by Chk1 (lines 386-401). This describes some evidence to suggest that radiosensitisation is evident in wild type p53 containing cell lines, including in isogenic derivatives, although this needs to be further explored. Just to note also that we previously acknowledged this evidence in the Conclusions and Future Directions section, and which does indeed have implications for the radiosensitisation of normal cells by Chk1 inhibition (lines 634-644).
- Previous preclinical evaluation of WEE1 or CHK1 inhibition as single-agent anticancer therapy (without irradiation) has shown large variation between different cancer cell lines. The differences in single-agent activity would be relevant also for the combined effects with irradiation, but have not been discussed.
Response: Whilst we appreciate the Reviewer’s suggestion, our review is obviously very much focussed on the radiosensitisation potential of Wee1 and Chk1, rather than their action as a monotherapy. Nevertheless, we have now added a paragraph into Section 4.5 on Clinical Trials which highlights the ongoing trials of Chk1/Wee1 inhibitors as single agent therapies, and which could inform on stratifying patients that could be responsive to the combination with radiation (lines 592-605).
- In line 582 it is mentioned that CHK1/WEE1 inhibitors with radiation lead to replication stress. Is this a specific effect for the combination with radiation? This issue could have been discussed more thoroughly. Many previous studies have shown that such inhibitors cause replication stress even in the absence of radiation. For instance, CHK1 or WEE1 inhibition causes aberrant replication initiation and subsequent DNA breakage in S-phase. Additionally, CHK1 and WEE1 inhibition may also protect against replication fork collapse
Response: We thank the Reviewer for their comment, although again would like to stress that our review is focussed on highlighting evidence of the combination of Wee1/Chk1 with radiation in causing phenotypes, such as replication stress. However, we have now referred to the fact that there is evidence that the inhibitors alone are capable of inducing replication stress (lines 514-519).
Reviewer 2 Report
Comments and Suggestions for Authors
The text for the following paragraph is very dense and difficult to follow. Breaking down into smaller paragraphs and more figures and graphs would be beneficial:
2. The cellular DNA damage response (DDR)
Similar comments for the following, we need to break down into a less dense text:
3. Roles of Chk1 and Wee1
When it comes to: 4.1. Development and progression of Chk1/Wee1 inhibitors, I would break it down to Chk1 separate to Wee1, as these are separate targetable areas.
Line 385: The PARP combination should be a separate section and highlighted as a promising suggestion.
Line 561: Repetition of previous statement
Line 608: 'Why would radiosensitising proton beam be different to photon beam, other than precision? Please explain'
Line 618: Do you have any specific trial proposals? This is a very interesting potential trial project.
Author Response
Reviewer 2 Comments and Suggestions for Authors
The text for the following paragraph is very dense and difficult to follow. Breaking down into smaller paragraphs and more figures and graphs would be beneficial:
- The cellular DNA damage response (DDR)
Response: We appreciate the Reviewer’s comment, and have now introduced two more breaks into Section 2 (lines 101 and 115) to make this more readable.
Similar comments for the following, we need to break down into a less dense text:
- Roles of Chk1 and Wee1
Response: As recommended by the Reviewer, breaks into Section 3 have now been added (lines 174 and 191).
When it comes to: 4.1. Development and progression of Chk1/Wee1 inhibitors, I would break it down to Chk1 separate to Wee1, as these are separate targetable areas.
Response: We thank the Reviewer for their suggestion, and the Development and Progression of Chk1 and Wee1 inhibitors has now been separated into Sections 4.1 and 4.2, respectively.
Line 385: The PARP combination should be a separate section and highlighted as a promising suggestion.
Response: We appreciate the Reviewer’s suggestion. However, the strategy of combining Chk1 inhibitors with PARP inhibitors was designed to highlight the importance of homologous recombination in the radiosensitisation of cells, rather than this being utilised as a potential therapeutic approach. Therefore, we have left this paragraph in its current position.
Line 561: Repetition of previous statement
Response: It is unclear from this point, what repetition the Reviewer is referring to.
Line 608: 'Why would radiosensitising proton beam be different to photon beam, other than precision? Please explain'
Response: We appreciate the Reviewer’s comment. As mentioned in Section 4.5, there is evidence to suggest that protons and carbon ions, the latter of which has a particularly higher LET than photon irradiation, induces a stronger G2/M arrest. With this higher activation of cell cycle arrest, it is feasible that Chk1 or Wee1 inhibitors may function more effectively as radiosensitisers with proton or carbon ion therapy that also generate more increased complex DNA damage. However, as mentioned in the Conclusions and Future Directions section (lines 663-671), this is contentious and needs exploring experimentally more thoroughly.
Line 618: Do you have any specific trial proposals? This is a very interesting potential trial project.
Response: We appreciate the Reviewer’s question. There still appears to be some lack of fundamental understanding of the mechanisms of cell death in response to the combination of Chk1/Wee1 inhibitors with radiation. Additionally, the specific tumours that would benefit from the treatment, as well as the specific radiation modalities (low or high-LET) that should be utilised are unclear. In our opinion, more preclinical research is needed in these areas before moving towards specific clinical trials.
Round 2
Reviewer 1 Report
Comments and Suggestions for Authors
N/A